# Molecular Pincers Using a Combination of N-H and C-H Donors for Anion Binding

**DOI:** 10.3390/ijms24010163

**Published:** 2022-12-22

**Authors:** Jaehyeon Kim, Seung Hyeon Kim, Nam Jung Heo, Benjamin P. Hay, Sung Kuk Kim

**Affiliations:** 1Department of Chemistry and Research Institute of Natural Science, Gyeongsang National University, Jinju 52828, Republic of Korea; 2Supramolecular Design Institute, Oak Ridge, TN 37830, USA

**Keywords:** molecular pincers, hydrogen bond, anion receptor, fluorescence, titration, association constant

## Abstract

A naphthalene imide (**1**) and a naphthalene (**2**) bearing two pyrrole units have been synthesized, respectively, as anion receptors. It was revealed by ^1^H NMR spectral studies carried out in CD_3_CN that receptors **1** and **2** bind various anions via hydrogen bonds using both C-H and N-H donors. Compared with receptor **2**, receptor **1** shows higher affinity for the test anions because of the enhanced acidity of its pyrrole NH and naphthalene CH hydrogens by the electron-withdrawing imide substituent. Molecular mechanics computations demonstrate that the receptors contact the halide anions via only one of the two respective available N-H and C-H donors whereas they use all four donors for binding of the oxyanions such as dihydrogen phosphate and hydrogen pyrophosphate. Receptor **1**, a push-pull conjugated system, displays a strong fluorescence centered at 625 nm, while receptor **2** exhibits an emission with a maximum peak at 408 nm. In contrast, upon exposure of receptors **1** and **2** to the anions in question, their fluorescence was noticeably quenched particularly with relatively basic anions including F^−^, H_2_PO_4_^−^, HP_2_O_7_^3−^, and HCO_3_^−^.

## 1. Introduction

Nature uses a hydrogen bond as one of the major driving forces for inter- or intramolecular interactions as well as anion recognition [1,2,3,4,5,6,7,8,9,10,11]. For instance, the helix structures of nucleic acids are constructed based on hydrogen bonding interactions between the base pairs of the respective nucleotide subunits [12,13,14,15]. Protein folding and protein–protein interactions are also mainly attributable to inter- or intra-molecular hydrogen bonds formed between the amide NH and CO groups within proteins [16,17,18,19,20]. In general, a hydrogen bond is a noncovalent interaction via electrostatic attraction between a hydrogen atom covalently bonded to a relatively large electronegative atom and another electronegative atom possessing a lone pair electron [1,2,3,4,5,6]. In supramolecular chemistry, hydrogen bonds have been widely utilized for anion recognition of a synthetic receptor [6,7,8,9]. This is because most anions are electron-rich and have lone pair electrons, in turn, acting as a hydrogen bond acceptor [7,8,9,10]. Anions play crucial roles in various areas associated with biology, chemistry, and the environment [21,22,23,24,25,26]. Therefore, numerous efforts have been devoted to the development of an artificial anion receptor capable of binding a certain anion selectively [27,28,29,30]. For anion recognition, O-H groups of carboxylic acids or alcohols and N-H groups belonging to amide, urea, squaramide, pyrrole, or indole units have been most commonly used as hydrogen bond donors [31,32,33,34,35]. For instance, a calix [4] pyrrole, a tetrapyrrolic macrocyclic compound having pure hydrogen bond donors, recognizes anions using suitably preorganized four pyrrolic NHs [36,37,38,39,40,41]. By contrast, squaramide and urea having both hydrogen bond donors and acceptors form a self-assembled structure via inter-molecular hydrogen bonds while they bind oxyanions as well as the halide anions in the presence of such anions [42,43,44,45]. More recently, it was reported that relatively less acidic C-H hydrogens of a neutral molecule could function as a hydrogen bond donor for anion binding [46,47,48,49]. For instance, in 2013, Flood and coworkers reported that a cyanostar macrocycle bearing five cyanostilbene subunits had an ability to complex large anions, including BF_4_^−^, ClO_4_^−^, and PF_6_^−^ via pure C-H•••anion hydrogen bonds [50]. Later, the Stępień group showed that a hyperbolic molecular belt bound the chloride anion selectively based on aromatic C-H•••Cl^−^ hydrogen bonds [51]. In 2019, the Flood group also reported that a cage molecule consisting of six triazole groups and three benzene units formed a strong complex with the chloride anion using nine C-H hydrogens as hydrogen bond donors [52]. By contrast, Maeda and coworkers reported in 2007 that aryl-functionalized dipyrrolyldiketone BF_2_ complexes were able to bind various anions via a combination of C-H•••anion and N-H•••anion hydrogen bonding interactions [53]. As an extension of the study on such anion receptors, we synthesized anion receptors **1** and **2** in which two pyrrole groups are directly linked to the naphthalene imide and naphthalene group, respectively. It was found via ^1^H NMR spectroscopic analyses that this new family of anion receptors (**1** and **2**) recognize anions via hydrogen bonds utilizing the naphthalene CHs as well as the pyrrole NHs. Receptors **1** and **2** are highly fluorescent and can therefore act as fluorescence sensors for anions. As inferred from the molecular structure, a pull-push type of the conjugated system (**1**) consisting of electron-donating and electron-accepting groups exhibits an emission peak of a significantly long wavelength at 625 nm relative to receptor **2** displaying its emission at 408 nm [54,55,56,57]. Here, we report the synthesis of receptors **1** and **2** and their anion binding features in solution (Figure 1).

## 2. Results and Discussion

The synthetic procedures for receptors **1** and **2** are shown in Figure 1. Briefly, N-protected pyrrole-2-boronic acid (**3**), dibromonaphthalene imide (**4**), and naphthalene ditriflate (**5**) were prepared following the previously reported procedures, respectively [58,59,60,61]. The Suzuki coupling reactions of compound **3** with dibromo compound **4** and ditriflate compound **5** carried out in THF/H_2_O in the presence of Pd(PPh_3_)_4_ as a catalyst and K_2_CO_3_ as a base produced naphthalene analogues **6** and **7** bearing two N-protected pyrroles in 54% and 75% yields, respectively. Finally, the tert-butoxycarbonyl groups of the pyrrole subunits were removed by heating ethylene glycol solutions containing compounds **6** or **7** to 180 °C for 1 h to give receptors **1** and **2** in 75% and 35% yields, respectively. Receptors **1** and **2** were structurally characterized via general spectroscopic analyses including ^1^H and ^13^C NMR spectroscopy and high-resolution mass spectrometry (HRMS).

Initial evidence for receptors **1** and **2** being capable of binding anions was gleaned from molecular mechanics computations performed utilizing the Merk Force Field 94 model (MMFF94) in vacuo [62,63]. The energies for the lowest energy forms of each of the anions and the receptors were used to calculate the binding energies (ΔE) defined as E_(host-guest complex)_ − E_(host)_ − E_(guest)_. The energies of the most stable ion-free forms of the receptors were calculated to be 74.57 kcal/mol and 33.76 kcal/mol for **1** and **2**, respectively, whereas the intrinsic energies of the respective isolated anions were computed to be 0 kcal/mol for the fluoride and chloride anion, −235.00 kcal/mol for the dihydrogen phosphate anion, and −95.53 kcal/mol for the hydrogen pyrophosphate anion. For the lowest energy structure of receptor **1** obtained from gas phase global minimum, the two pyrrolic NHs were oriented to the imide carbonyl oxygen atoms, possibly to gain favorable electrostatics from C=O•••H-N orientations (Figure 2). By contrast, in the case of receptor **2**, its two pyrrole NHs were directed to the same orientation as the two naphthalene CHs located between the two pyrrole subunits. The respective pyrroles were also found to face below and above the naphthalene plane (Figure 2). These receptor architectures can form a cavity with two N-H and two C-H hydrogen bond donors oriented in the same direction.

The binding energies (ΔE) for the fluoride and the chloride complex of receptor **1** were computed to be −28.08 kcal/mol and −23.58 kcal/mol, respectively, while receptor **2** was found to bind the fluoride and the chloride anion with −22.61 kcal/mol and −18.46 kcal/mol of binding energies, respectively (Figure 3). These values reflect that receptor **1** binds the fluoride and the chloride anion more strongly by 5.47 kcal/mol and 5.12 kcal/mol, respectively, than receptor **2**. This finding is attributable to the high acidity of the pyrrolic NHs and naphthalene CHs of receptor **1** bearing the electron-withdrawing imide group relative to receptor **2**. This suggestion was further supported by the higher affinity of receptor **1** for anions than that of receptor **2** as inferred from ^1^H NMR spectroscopic studies (vide infra). The optimized structures of the fluoride and the chloride complexes of receptors **1** and **2** showed that both receptors bind the fluoride and the chloride anion with 1:1 stoichiometry and that both halide anions can only contact one of the N-H + C-H chelate rings of the receptors via hydrogen bonds (Figure 3). Thus, although receptors **1** and **2** have four hydrogen bond donors, the receptor architecture prevents all four of these donor groups from simultaneously contacting these mono-atomic halide anions. That is, the fluoride and chloride anions are too small to bind all four of these donors simultaneously. Due to such structural constraints, these halide anions can only interact with 1/2 of the available binding sites in the receptor cavity. By contrast, receptors **1** and **2** complex the dihydrogen phosphate anion via a different binding mode from the chloride anion. For instance, the computed lowest energy conformations of the dihydrogen phosphate complexes of receptors **1** and **2** reveal that this oxyanion is able to contact both N-H + C-H chelate rings. Structurally, the geometry of receptors **1** and **2** is better organized for binding two adjacent O atoms in the dihydrogen phosphate than for binding a halide anion (Figure 3). The binding energies of the receptors for the dihydrogen phosphate anion were computed to be −37.96 kcal/mol and −31.54 kcal/mol for **1** and **2**, respectively, a finding mirroring that receptor **1** forms a more stable complex with the dihydrogen phosphate anion by 6.42 kcal/mol than receptor **2**. We also computed the lowest energy structures and the binding energies for the hydrogen pyrophosphate complexes of receptors **1** and **2**. In analogy to what was seen with dihydrogen phosphate, mono-hydrogen pyrophosphate (HP_2_O_7_^3−^) formed hydrogen bonds to both pyrrole N-H and naphthalene C-H hydrogens of the receptors. In these cases, one P-O group from each phosphate serves as the hydrogen bond acceptor (Figure 3). The resulting binding energies were calculated to be −71.69 kcal/mol and −55.76 kcal/mol for receptors **1** and **2**, respectively, which suggests that receptor **1** also forms a stronger complex with the hydrogen pyrophosphate anion by 15.93 kcal/mol than **2**. Taken together, it could be concluded from the computations that receptor **1** binds the anions in question more favorably than receptor **2** and that both receptors form stronger complexes with the dihydrogen phosphate and hydrogen pyrophosphate anions than with the halide anions. This conclusion is consistent with the results obtained from ^1^H NMR spectral studies carried out in CD_3_CN (vide infra).

Receptors **1** and **2** were evaluated via ^1^H NMR spectroscopic analyses in terms of their ability to bind anions in CD_3_CN solutions. For instance, when receptor **1** was treated with representative anions including F^−^, Cl^−^, Br^−^, I^−^, HSO_4_^−^, H_2_PO_4_^−^, and HP_2_O_7_^3−^ (as their tetrabutylammonium (TBA^+^) salts) as well as HCO_3_^−^ (as its tetraethylammonium (TEA^+^) salt) in CD_3_CN, most anions gave rise to chemical shift changes for the proton signals of receptor **1** consistent with anion binding by the receptor. Specifically, upon exposure of receptor **1** to those anions, the proton signals corresponding to the pyrrolic NHs and the naphthalene aromatic CHs (H_d_) experienced noticeable shifts to downfield while the pyrrole CH proton resonances were slightly upfield-shifted in ^1^H NMR spectra (Figure 4). In particular, in the presence of fluoride, dihydrogen phosphate, hydrogen pyrophosphate, and bicarbonate, both proton signals of the NHs and the aromatic CHs (H_d_) underwent large downfield shifts. Specifically, upon addition of the dihydrogen phosphate anion to receptor **1** in CD_3_CN, the receptor NH and CH (H_d_) proton signals originally appearing at δ = 10.02 ppm and 8.44 ppm, respectively, were downfield shifted to δ = 13.47 ppm (Δδ = 3.45 ppm) and 9.13 ppm (Δδ = 0.69 ppm), respectively (Figure 4). In agreement with what was inferred from the molecular mechanics computations, these ^1^H NMR spectral changes were taken as critical evidence that the naphthalene CH hydrogens as well as the pyrrole NH hydrogens participate in anion binding via hydrogen bonds (Figure 3). In analogy to what was seen with receptor **1**, receptor **2** was also found to undergo noticeable chemical shift movements in its proton signals when exposed to the test anions in CD_3_CN. This finding also suggests that receptor **2** recognizes the anions via a combination of C-H•••anion and N-H•••anion hydrogen bonds (Figure 5).

Detailed ^1^H NMR spectroscopic titrations of receptors **1** and **2** with the test anions were carried out in CD_3_CN to quantify their binding affinity for the anions under study. For instance, upon the subjection of receptor **1** to titration with the fluoride anion, the proton signals corresponding to the pyrrole NHs and the naphthalene imide CHs (H_d_) were dramatically downfield-shifted with increasing quantities of the fluoride anion (Figure 6). These chemical shift changes were attributable to the hydrogen bonds formed between the fluoride anion and both pyrrole NH and aromatic CH protons of receptor **1**. These interactions also served to increase the electron density of the naphthalene ring leading to an appreciable upfield shift of the other proton signal corresponding to the naphthalene CHs (H_c_). The resulting binding isotherm was best fitted to a standard 1:1 binding profile leading to the association constant (K_a_) of 1653 M^−1^ [64]. The presumed 1:1 binding stoichiometry of receptor **1** for fluoride was further confirmed by Job’s plot experiment (Figure 7). We also carried out titration experiments with poly atomic oxyanions (including HSO_4_^−^, H_2_PO_4_^−^, HP_2_O_7_^3−^, and HCO_3_^−^) bearing multi-hydrogen bond-accepting oxygen atoms. Specifically, when receptor **1** was titrated with the dihydrogen phosphate anion in CD_3_CN, gradual downfield shifts of both singlet proton signals corresponding to the pyrrole NHs and the naphthalene CHs, respectively, took place before saturation was reached upon addition of 5.34 anion equiv (Figure 8). From this titration and a Job’s plot experiment, receptor **1** was found to bind the dihydrogen phosphate anion with an association constant of K_a_ = 1326 M^−1^ and with 1:1 stoichiometry (Figure 7 and Figure 8). Analogous chemical shift changes in the receptor proton signals but with different extents were observed upon exposure of receptor **1** to incremental amounts of other anions such as Cl^−^, Br^−^, I^−^, HSO_4_^−^, HP_2_O_7_^3−^, and HCO_3_^−^ (Appendix A). These titration experiments allowed us to estimate the association constants (K_a_) of receptor **1** for the anions to be 157 M^−1^ for chloride, 25 M^−1^ for bromide, 3 M^−1^ for iodide, 28 M^−1^ for hydrogen sulfate, 1531 M^−1^ for hydrogen pyrophosphate, and 2269 M^−1^ for bicarbonate, respectively (Table 1) [64]. Receptor **2** showed similar chemical shift changes of the singlet proton signals associated with the pyrrole NH and the naphthalene CH (H_a_) when titrated with the anions in CD_3_CN. From these titration experiments, the association constants of receptor **2** for the anions were calculated to be 196 M^−1^ for fluoride, 43 M^−1^ for chloride, 13 M^−1^ for bromide, 2 M^−1^ for iodide, 1234 M^−1^ for hydrogen sulfate, 771 M^−1^ for hydrogen pyrophosphate, and 946 M^−1^ for bicarbonate, respectively (Table 1; Appendix A). As shown in Table 1, the association constants of receptor **2** for the anions in question were considerably small as compared with those of receptor **1**. Again, this finding is attributable to relatively enhanced acidity of the pyrrole NH and naphthalene CH of receptor **1** presumably because of the electron-withdrawing imide group.

Because receptors **1** and **2** exhibited strong fluorescence, we examined their optical properties using fluorescence in the absence and presence of the anions. As expected from the molecular structure, receptor **1**, an electron pull-push conjugated system consisting of two electron-donating pyrrole groups directly linked to an electron-accepting the naphthalene imide unit, displays an absorption peak and an emission peak at significantly longer wavelengths than receptor **2** [54,55,56,57]. For instance, receptor **1** exhibits its absorption peak at 448 nm and its emission peak at 625 nm upon excitation at 448 nm, respectively, while receptor **2** displays its maximum absorption and emission peak at 307 nm and 408 nm, respectively (Figure 9). When receptor **1** was treated with excess amounts of the respective anions, its fluorescence was quenched by most test anions. Especially, basic anions such as F^−^, H_2_PO_4_^−^, HP_2_O_7_^3−^, and HCO_3_^−^ were found to give rise to significant fluorescence quenching (Figure 10a). In contrast, relatively small levels of fluorescence quenching were observed upon exposure of receptor **2** to most test anions (Figure 10b). By contrast, the fluoride anion led receptor **2** to emit red-shifted fluorescence at 498 nm with weakened intensity accompanying a color change from sky blue to cyan (Figure 10b). This fluorescence wavelength change is presumably attributed to partial deprotonation of the pyrrole NH protons of receptor **2** in the presence of an excess quantity of the fluoride anion. We also carried out quantitative titration experiments of receptors **1** and **2** with the anions using fluorescence spectroscopy to evaluate their anion binding affinity at a low concentration relative to the ^1^H NMR spectral titration condition. For instance, when receptor **1** was subjected to titrations with fluoride, dihydrogen phosphate, hydrogen pyrophosphate, and bicarbonate, respectively, its fluorescence displayed at 625 nm was gradually quenched with slight hypsochromic shifts at high concentrations of those anions (Figure 11). By contrast, other anions gave rise to relatively small fluorescence quenching with no wavelength changes. From these fluorescence titrations, the association constants of receptor **1** were estimated to be 8493 M^−1^ for fluoride, 484 M^−1^ for chloride, 674 M^−1^ for bromide, 1996 M^−1^ for iodide, 1976 M^−1^ for hydrogen sulfate, 101,360 M^−1^ for dihydrogen phosphate, 306,690 M^−1^ for hydrogen pyrophosphate, and 11,911 M^−1^ for bicarbonate (Figure 11, Appendix A) [64]. Receptor **2** was also found to exhibit steadily decreasing fluorescence at 448 nm but without wavelength changes over the course of its titrations with the anions in question (Appendix A). Unfortunately, we failed to obtain reliable association constants of receptor **2** for the anions from these fluorescence spectral titrations because error ranges are too large for most anions.

## 3. Materials and Methods

### 3.1. General Experimental

Solvents and reagents used for the synthetic work were purchased from Aldrich, TCI, Acros Organics or Alfa Aesar and used without further purification. Compounds **3**–**5** were prepared as reported previously [58,59,60,61]. NMR spectra were recorded on a Bruker Advance-300 MHz instrument. The NMR spectra were referenced to residual solvent peaks and the spectroscopic solvents were purchased from either Cambridge Isotope Laboratories or Deutero GmbH Laboratories. The Q-TOF HRMS data was recorded on a Waters (Xevo G2-XS Tof) instrument. UV/Vis spectra were measured on a Mega-800 (SCINCO) spectrometer. Fluorescence spectra were measured on a RF-6000 (SHIMADZU) spectrometer. TLC analyses were carried out using Sorbent Technologies silica gel (200 mm) sheets. Column chromatography was performed on Sorbent silica gel 60 (40–63 mm). For computations, PCModel was used to locate the lowest energy conformation for the free ligand and various ligand anion complexes [62]. This was done by sampling structures taken from 50 ps molecular dynamics runs using the Merck Force Field 94 model in vacuo [63]. The energies for the lowest energy forms of each species were used to compute interaction energies:∆E = E_(complex)_ − E_(host)_ − E_(guest)_

### 3.2. Synthesis of Compound **6**

A mixture of THF (100 mL) and water (20 mL) was stirred and degassed by bubbling with nitrogen gas for 1 h. Compound **5** (1.00 g, 2.00 mmol), N-Boc-pyrole-2-boronic acid **3** (1.23 g, 6.00 mmol), Pd(PPh_3_)_4_ (0.23 g, 0.20 mmol), and K_2_CO_3_ (2.68 g, 20 mmol) were added to the degassed solvent mixture. The reaction mixture was stirred and heated to reflux for 18 h. The relatively volatile THF solvent was removed under vacuum. The resulting aqueous solution containing the product was extracted with CH_2_Cl_2_ (2 × 50 mL). The combined organic layers were washed twice with distilled water and dried over magnesium sulfate. The solvent was evaporated under reduced pressure. Purification of the crude product by column chromatography over silica gel (ethyl acetate/hexanes = 1/5) afforded the product **6** (0.72 g, 54% yield) as a yellowish solid. ^1^H NMR (300 MHz, Chloroform-d): δ 8.62 (d, J = 1.6 Hz, 2H), 8.26 (d, J = 1.6 Hz, 2H), 7.53–7.42 (m, 3H), 7.33 (d, J = 7.7 Hz, 2H), 6.43 (dd, J = 3.4, 1.8 Hz, 2H), 6.32 (t, J = 3.3 Hz, 2H), 2.74 (p, J = 6.8 Hz, 2H) 1.36 (s, 18H), 1.15 (d, J = 6.8 Hz, 12H). ^13^C NMR (126 MHz, Chloroform-d): δ 164.2, 149.2, 145.5, 133.7, 133.5, 132.8, 132.3, 131.6, 130.7, 129.6, 126.4, 124.1, 124.0, 121.5, 116.2, 111.1, 84.6, 29.2, 27.6, 24.0. HR FAB-MS m/z 688.3387 [M+H]^+^ calcd for C_42_H_46_N_3_O_6_, found 688.3362.

### 3.3. Synthesis of Compound **7**

A mixture of THF (100 mL) and water (20 mL) was degassed by bubbling with nitrogen gas for 1 h. Compound **6** (1.00 g, 2.40 mmol), N-Boc-pyrole-2-boronic acid (1.23 g, 7.10 mmol), Pd(PPh_3_)_4_ (0.27 g, 0.24 mmol), and K_2_CO_3_ (2.68 g, 24 mmol) were added to the mixed solvents. The mixture was stirred at reflux for 18 h. The organic solvents were evaporated in vacuo. The resulting aqueous solution was extracted with CH_2_Cl_2_ (2 × 50 mL). The organic layers were collected and washed twice with brine and then with water. The combined organic layer was dried over MgSO_4_, and the solvents were removed under reduced pressure. Purification of the crude product by column chromatography (CH_2_Cl_2_/hexanes = 1/4 to 1/1) over silica gel gave compound **7** (0.70 g, 65% yield) as sticky colorless liquid. ^1^H NMR (300 MHz, Chloroform-d): δ 7.84–7.74 (m, 4H), 7.45 (dd, J = 8.4 and 1.3 Hz, 2H), 7.40 (dd, J = 3.2 and 1.9 Hz, 2H), 6.35–6.22 (m, 4H), 1.32 (s, 18H). ^13^C NMR (126 MHz, DMSO-d_6_): δ 149.4, 134.7, 132.1, 128.1, 127.2, 127.0, 123.4, 115.4, 111.6, 84.3, 55.2, 27.7, 27.5. HR FAB-MS m/z 458.2206 [M]^+^ calcd for C_28_H_30_N_2_O_4_, found 458.2200.

### 3.4. Synthesis of Compound **1**

A solution of compound **6** (0.60 g, 0.90 mmol) in ethylene glycol (5 mL) was heated to 180 ^o^C and stirred for 1 h at this temperature. After cooling the reaction mixture, water (100 mL) and methylene chloride (100 mL) were added to the mixture. The organic layer was separated and washed twice with water and then dried over MgSO_4_. After filtration, the organic solvent was evaporated under reduced pressure. The resulting crude product was purified by column chromatography (ethyl acetate/hexanes = 1/4 to 1/2) to afford receptor **1** (0.32 g, 75% yield) as a red powder solid. ^1^H NMR (300 MHz, DMSO-d_6_): δ 11.84 (s, 2H), 8.70 (d, J = 1.7 Hz, 2H), 8.58 (d, J = 1.7 Hz, 2H), 7.47 (dd, J = 8.5 and 6.9 Hz, 1H), 7.36 (d, J = 7.3 Hz, 2H), 7.01 (td, J = 2.7 and 1.5 Hz, 2H), 6.86 (dq, J = 3.7 and 1.6 Hz, 2H), 6.24 (dt, J = 3.6 and 2.3 Hz, 2H), 2.73 (p, J = 6.8 Hz, 2H), 1.08 (d, J = 6.8 Hz, 12H). ^13^C NMR (126 MHz, DMSO-d_6_): δ 164.2, 146.0, 133.7, 132.9, 131.6, 130.0, 129.6, 126.8, 126.6, 124.7, 124.2, 122.8, 121.7, 110.4, 108.3, 29.0, 24.1. HR FAB-MS m/z 488.2338 [M+H]^+^ calcd for C_32_H_29_N_3_O_2_, found 488.2333.

### 3.5. Synthesis of Compound **2**

An ethylene glycol solution of compound **7** (0.70 g, 0.90 mmol) was heated and stirred at 180 ^o^C for 1 h. The organic solvents were removed under vacuum. After cooling the reaction mixture, water (100 mL) and methylene chloride (100 mL) were added to the mixture. The organic layer was separated and washed twice with water and then dried over MgSO_4_. After filtration, the organic solvent was evaporated under reduced pressure. The resulting crude mixture was purified by column chromatography over silica gel (eluent: ethyl acetate/hexanes = 1/4 to 1/1) to give receptor **2** (0.14 g, 35% yield) as a gray solid. ^1^H NMR (300 MHz, DMSO-d_6_): δ 11.47 (s, 2H), 8.01 (d, J = 1.7 Hz, 2H), 7.81 (d, J = 8.6 Hz, 2H), 7.72 (dd, J = 8.5 and 1.6 Hz, 2H), 6.91 (q, J = 2.3 Hz, 2H), 6.65 (m, 2H), 6.17 (q, J = 2.7 Hz, 2H). ^13^C NMR (126 MHz, DMSO-d_6_): δ 134.4, 131.6, 131.4, 130.1, 128.4, 122.7, 120.4, 120.3, 109.8, 106.8. HR FAB-MS m/z 259.1235 [M+H]^+^ calcd for C_18_H_14_N_2_, found 259.1212.

## 4. Conclusions

In conclusion, molecular pincers **1** and **2**, consisting of two pyrroles directly linked to a naphthalene imide and the plain naphthalene, respectively, were synthesized as anion receptors. These compounds have architectures capable of forming a cavity with two N-H and two C-H hydrogen bond donors facing in the same general direction. ^1^H NMR spectroscopic analyses demonstrate that both receptors (**1** and **2**) bind the test anions using both the pyrrole N-H and naphthalene C-H hydrogen bond donors. It was proposed via molecular mechanics computations that the receptors complex the anions via different binding modes depending on the size and shape of the anions. For instance, due to structural constraints, the receptors bind the halide anions such as fluoride (F^–^) and chloride (Cl^–^) using only one of two available N-H and C-H hydrogen bond donors. In contrast, the dihydrogen phosphate (H_2_PO_4_^–^) and the hydrogen pyrophosphate (HP_2_O_7_^3–^) anion are bound to all four donors simultaneously but with a different motif. As proved by the association constants (K_a_) estimated from ^1^H NMR spectral titrations carried out in CD_3_CN, receptor **1** binds most anions with high affinity as compared with receptor **2**. This is attributable to the inherently high acidity of the pyrrole NHs and the naphthalene CHs of receptor **1** relative to **2** because of the electron-withdrawing imide group in **1**. An electron push-pull conjugated system, receptor **1** exhibits strong fluorescence at 625 nm, which is a much longer wavelength than the fluorescence emitted by receptor **2**. When receptors **1** and **2** were exposed to the anions in question, they displayed noticeably quenched fluorescence. In particular, fluoride, bicarbonate, dihydrogen phosphate, and hydrogen pyrophosphate gave rise to a large level of fluorescence changes for receptor **1**.

## Data Availability

Not applicable.

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
