# Peer review of "Molecular Pincers Using a Combination of N-H and C-H Donors for Anion Binding"

_ijms, 2022, doi:10.3390/ijms24010163_

Round 1
Reviewer 1 Report
In this manuscript, Kim et.al synthesized two naphthalene-based anion receptors. One contains a phenyl imide structure, the other does not. Both receptors show the capability of binding a series of anions through NHs and CHs hydrogen bond donors, while the former in general exhibits higher association constants. The authors predicted the binding position using molecular mechanics and confirmed it with 1H NMR. The photophysical properties of these anion receptors show the potential of being used as fluorescence probes for different anions. The overall quality of this manuscript is good, and I would recommend it for publication after minor revision. Here are several comments below that would be helpful:
1. In lines 96 - 97, “possibly to gain favorable electrostatics from C=O•••H-N orientations”. I am not sure this interaction is possible, as the C=O group of imide is far from the N-H of Boc in space. Can the authors show any evidence or example of such interaction?
2. In lines 259 – 260, why is the error too large to obtain the association constant for 2? What is the large error? And where is it coming from?
3. In figure 9, the legend is not correct. The orange curves should be 1.
4. I wonder if the binding would influence the absorption of these receptors. If so the author should use the wavelength at the isosbestic point for excitation.
Reviewer 2 Report
It is valuable work, but its weakness is the lack of a solid quantum description of structures and stabilities of the studied systems. The application of the molecular mechanics method with the Merk Force Field 94 87 (MMFF94) is rather insufficient. For some of the studied systems with strong hydrogen bonds, the potential for the proton motion may have a bistable character. This, in some cases, may cause a strong shift of the protons towards the anions. Such a phenomenon cannot be described using a conventional force field which assumes a fixed topology of the chemical bonds.
I would suggest that the Authors perform quantum mechanical calculations for the studied systems with geometry optimisation, e.g. at the DFT B3LYP/6-311++G(d,p) or M06-2X/6-311++G(d,p) levels. The MMFF94 optimised geometries reported in the supplementary material can be used as starting points for the quantum-mechanical calculations.
It may also be worth referring to the following review: S. Scheiner, The Hydrogen Bond: A Hundred Years and Counting, J. Indian Inst. of Science. 2019, 100, 1-27, and some reference cited in it.
